# Local Linear Scale Factors in Map Projections of an Ellipsoid

Miljenko Lapaine 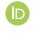

Faculty of Geodesy, University of Zagreb, Kačićeva 26, 10000 Zagreb, Croatia; mlapaine@geof.hr

**Abstract:** The main problem in cartography is that it is not possible to map/project/transform a spherical or ellipsoidal surface into a plane without distortions. The distortions of areas, angles, and/or distances are immanent to all maps. It is known that scale changes from point to point, and at certain points, the scale usually depends on the direction. The local linear scale factor $c$ is one of the most important indicators of distortion distribution in the theory of map projections. It is not possible to find out the values of the local linear scale factor $c$ in directions of coordinate axes $x$ and $y$ immediately from the definition of $c$. To solve this problem, in this paper, we derive new formulae for the calculation of $c$ for a rotational ellipsoid. In addition, we derive the formula for computing $c$ in any direction defined by $dx$ and $dy$. We also considered the position and magnitude of the extreme values of $c$ and derived new formulae for a rotational ellipsoid.

**Keywords:** map projection; scale factor; distortion distribution





## 1. Introduction

A map is a result of mapping data, usually from the Earth, a celestial body, or an imagined world to a plane of representation on a piece of paper or on a digital display such as a computer monitor. Usually, maps are created by transforming data to a spherical or ellipsoidal surface and then to a plane. The mapping from a curved surface into a plane is known as *map projection* and can take a variety of forms [1].

Since no map projection maintains the correct scale throughout, it is important to determine the extent to which it varies on a map. On a world map, qualitative distortion is evident to an eye familiar with maps, after noting the extent to which landmasses are improperly sized or out of shape, and the extent to which meridians and parallels do not intersect at right angles or are not spaced uniformly along a given meridian or given parallel. On maps of countries or even of continents, distortion may not be evident to the eye, but it becomes apparent upon careful measurement and analysis [2].

All map projections involve the distortion of areas, angles, and/or distances. The types of distortion can be controlled to preserve specific characteristics, but map projections must distort other characteristics of the represented object. The main problem in cartography is that it is not possible to map a spherical or ellipsoidal surface into a plane without distortions. Euler first proved as early as 1772 that a sphere cannot be mapped into a plane with zero distortion [3–5].

The principal (linear) scale PS is the ratio of the length in the plane of projection and its origin on the surface (sphere, ellipsoid) to be projected/mapped. Notice that the scale of a map is not the ratio of a distance on the map to the corresponding distance on the ground [6].

The PS is usually indicated on maps because it determines the general degree of reduction in the length on the map. On most maps, it is usually simply called 'scale' and is known as the map scale.

The scale changes from point to point, and usually depends on direction. This is the local scale. The *local linear scale factor $c$* is the ratio of the differential of the curve arc in the plane of projection and the differential of the corresponding curve arc on an ellipsoid or spherical surface (see details in Section 2).

Tissot's indicatrix is often used to illustrate the variation of point scale across a map. Each function can be locally approximated by a linear function. Each cartographic projection can be locally approximated by affine mapping. This was noticed by N. Tissot in the second half of the 19th century [7]. By mapping a circle of infinitesimal radius, an ellipse is obtained, which is called the distortion ellipse or Tissot's indicatrix. The directions of this ellipse are called the main directions, and along them, the local linear scale is both the largest and the smallest.

A single indicatrix describes the distortion at a single point. Because distortion varies across a map, generally, Tissot's indicatrices are placed across a map to illustrate the spatial change in distortion. A common scheme places them at each intersection of the displayed meridians and parallels (Figure 1). These schematics are important in the study of map projections, both to illustrate distortion and to provide the basis for the calculations that represent the magnitude of distortion precisely at each point.

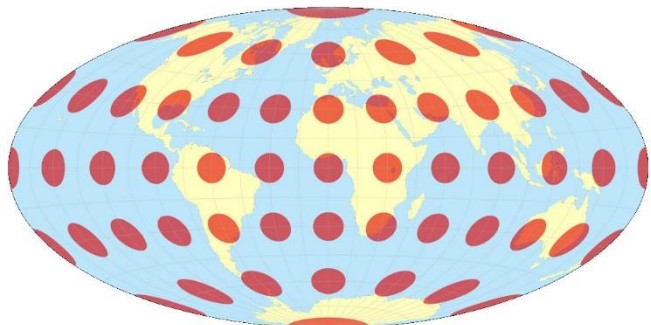

**Figure 1.** The Hammer projection with Tissot's indicatrices.

Tissot's indicatrices illustrate linear, angular, and areal distortions of maps, defined as follows:

- A map distorts distances (linear distortion) wherever the quotient between the lengths of an infinitesimally short line as projected onto the projection surface and as it originally is on the Earth model deviates from 1.
- A map distorts angles wherever the angles measured on the model of the Earth are not conserved in the projection. This is expressed by an ellipse of distortion, which is not a circle.
- A map distorts areas wherever areas measured in the model of the Earth are not conserved in the projection. This is expressed by ellipses of distortion whose areas vary across the map.

In conformal maps, where each point preserves angles projected from the geometric model, the Tissot's indicatrices are all circles whose size varies by location. In equal-area projections, where area proportions between objects are conserved, the Tissot's indicatrices all have the same area, though their shapes and orientations vary with location. In arbitrary projections, both the area and shape vary across the map.

Since it is not possible to read a local linear scale factor in the direction of a coordinate axis immediately from the definition of a local linear scale factor, this paper considers the derivation of new formulae that enable local linear scale factors in the direction of coordinate $x$ and $y$ axes to be calculated for mapping a rotational ellipsoid. The formula for computing the local linear scale factor in any direction defined by $dx$ and $dy$ is also derived. Furthermore, the position and magnitude of the extreme values of the local linear scale factor are considered and new formulas are derived. The paper is a generalization to the ellipsoid of a previous paper developed for a sphere [8].

## 2. Ellipsoid, Map Projection, and Local Linear Scale Factor

This chapter does not contain novelties in the theory of cartographic projections, but it is necessary for an understanding of what follows. The geodetic parameterization of

a rotational ellipsoid with semiaxes *a* and *b* and the center located in the origin of the coordinate system is a mapping defined by the following formulae [2]:

$$x = N \cos \varphi \cos \lambda, \ y = N \cos \varphi \sin \lambda, \ z = N\left(1 - e^2\right) \sin \varphi \tag{1}$$

where:

$$e^2 = \frac{a^2 - b^2}{a^2} \tag{2}$$

$$N = \frac{a}{\sqrt{1 - e^2 \sin^2 \varphi}}, \tag{3}$$

$\varphi \in \left[-\frac{\pi}{2}, \frac{\pi}{2}\right], \lambda \in [-\pi, \pi]$. In this case, $\varphi$ is the latitude, and $\lambda$ is the longitude. It is not difficult to derive that the first fundamental form of this mapping reads as:

$$ds^2 = M^2 d\varphi^2 + N^2 \cos^2 \varphi d\lambda^2 \tag{4}$$

where:

$$M = \frac{a\left(1 - e^2\right)}{\sqrt{\left(1 - e^2 \sin^2 \varphi\right)^3}} \tag{5}$$

A map projection is mapping given by the functions:

$$x = x(\varphi, \lambda), \ y = y(\varphi, \lambda), \tag{6}$$

where the geodetic coordinates are $\varphi \in \left[-\frac{\pi}{2}, \frac{\pi}{2}\right], \lambda \in [-\pi, \pi]$, as usual, and *x* and *y* are the coordinates of a point in a rectangular (mathematical, right-oriented) coordinate system in a plane. The first fundamental form of such a mapping is [9]:

$$ds'^2 = Ed\varphi^2 + 2Fd\varphi d\lambda + Gd\lambda^2, \tag{7}$$

with the coefficients:

$$E = \left(\frac{\partial x}{\partial \varphi}\right)^2 + \left(\frac{\partial y}{\partial \varphi}\right)^2, \ F = \frac{\partial x}{\partial \varphi}\frac{\partial x}{\partial \lambda} + \frac{\partial y}{\partial \varphi}\frac{\partial y}{\partial \lambda}, \ G = \left(\frac{\partial x}{\partial \lambda}\right)^2 + \left(\frac{\partial y}{\partial \lambda}\right)^2. \tag{8}$$

Figure 2 on the left-hand side shows a differential quadrangel ABCD on the ellipsoid, while on the right-hand side is the image A′B′C′D′ of that quadrangle in the plane of projection. The following labels were used:

*ds*: differential arc on the ellipsoid surface;
*ds′*: image of ds in the plane of projection;
*α*: the angle between the differential arc on the ellipsoid surface and a meridian;
*α′*: image of *α*;
θ: the angle between the images of a meridian and a parallel.

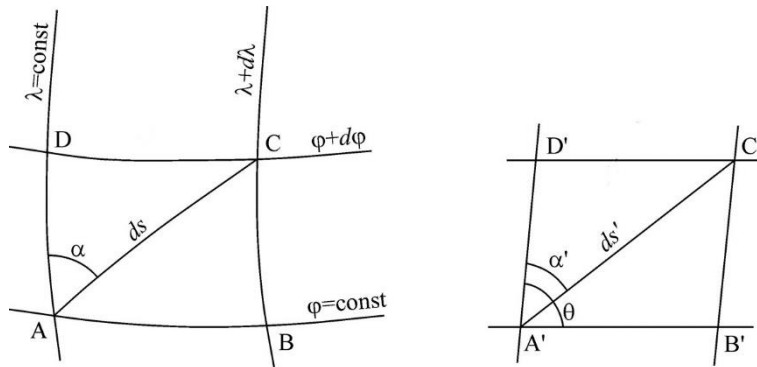

**Figure 2.** An ellipsoidal differential quadrangle (**left**) and its image in the plane of projection (**right**).

The local linear scale factor $c$ for mapping (3) of a rotational ellipsoid is usually defined in the theory of map projections by using the following relation:

$$c^2 = \frac{ds\prime^2}{ds^2} = \frac{E d\varphi^2 + 2F d\varphi d\lambda + G d\lambda^2}{M^2 d\varphi^2 + N^2 \cos^2 \varphi d\lambda^2} \tag{9}$$

which can also be written as follows in [2,3]:

$$c^2(\alpha) = \frac{E}{M^2} \cos^2 \alpha + \frac{F}{MN \cos \varphi} \sin 2\alpha + \frac{G}{N^2 \cos^2 \varphi} \sin^2 \alpha \tag{10}$$

where:

$$\tan \alpha = \frac{N \cos \varphi d\lambda}{M d\varphi} \tag{11}$$

The poles are singular points of geographic parameterization (1), and therefore, expression (9) and all subsequent ones should be interpreted in the poles as limiting cases when $\varphi \to \frac{\pi}{2}$ or $\varphi \to -\frac{\pi}{2}$.

If $\alpha = 0$ or, more generally, $\alpha = z\pi$, $z \in Z$, where $Z$ denotes the set of all integers, then the local linear scale factor $c$ along a meridian ($d\lambda = 0$) is:

$$h = c(d\lambda = 0) = \frac{\sqrt{E}}{M} \tag{12}$$

and if $\alpha = \frac{\pi}{2}$ or, more generally, $\alpha = \frac{\pi}{2} + z\pi$, $z \in Z$, then the local linear scale factor $c$ along a parallel ($d\varphi = 0$) is given by:

$$k = c(d\varphi = 0) = \frac{\sqrt{G}}{N \cos \varphi}. \tag{13}$$

## 3. Local Linear Scale Factors in the Directions of Coordinate Axes

It is not possible to read a local linear scale factor in the direction of a coordinate axis immediately from the definition of local linear scale factor (9). The same is true for Equation (10), where $\alpha$ denotes the azimuth, i.e., the angle between a meridian and any direction in a point in question. To be able to obtain a local linear scale factor in a direction defined by $dx$ and $dy$, we need to modify Equation (9) or (10) in the appropriate way.

Let us start with the general Equations (6) of a map projection. Then, we can write the following:

$$dx = \frac{\partial x}{\partial \varphi} d\varphi + \frac{\partial x}{\partial \lambda} d\lambda, \; dy = \frac{\partial y}{\partial \varphi} d\varphi + \frac{\partial y}{\partial \lambda} d\lambda. \tag{14}$$

From (14), we have:

$$d\varphi = -\frac{1}{H} \left( \frac{\partial y}{\partial \lambda} dx - \frac{\partial x}{\partial \lambda} dy \right), \; d\lambda = \frac{1}{H} \left( \frac{\partial y}{\partial \varphi} dx - \frac{\partial x}{\partial \varphi} dy \right), \tag{15}$$

where:

$$H = \sqrt{EG - F^2} = \left| \frac{\partial y}{\partial \varphi} \frac{\partial x}{\partial \lambda} - \frac{\partial y}{\partial \lambda} \frac{\partial x}{\partial \varphi} \right| \tag{16}$$

and

$$H > 0. \tag{17}$$

If we suppose that:

$$dy = 0 \tag{18}$$

then:

$$d\varphi = -\frac{1}{H} \frac{\partial y}{\partial \lambda} dx, \; d\lambda = \frac{1}{H} \frac{\partial y}{\partial \varphi} dx, \tag{19}$$

and by substituting (19) in (9), we obtain the local linear scale factor in the direction of the *x*-axis as follows:

$$c(dy = 0) = \frac{H}{\sqrt{M^2 \left(\frac{\partial y}{\partial \lambda}\right)^2 + N^2 \cos^2 \varphi \left(\frac{\partial y}{\partial \varphi}\right)^2}} \tag{20}$$

If we suppose that:

$$dx = 0 \tag{21}$$

then:

$$d\varphi = \frac{1}{H} \frac{\partial x}{\partial \lambda} dy, \; d\lambda = -\frac{1}{H} \frac{\partial x}{\partial \varphi} dy, \tag{22}$$

and by substituting (19) in (9), we obtain the local linear scale factor in the direction of the *y*-axis as follows:

$$c(dx = 0) = \frac{H}{\sqrt{M^2 \left(\frac{\partial x}{\partial \lambda}\right)^2 + N^2 \cos^2 \varphi \left(\frac{\partial x}{\partial \varphi}\right)^2}} \tag{23}$$

From the general theory of map projections, it is known that the local linear scale factor can be visualized as an ellipse. It is the ellipse of distortion or the Tissot's indicatrix. Figure 3 represents a general case of Tissot's indicatrix with local linear scale factors along a meridian $c(d\lambda = 0)$, along a parallel $c(d\varphi = 0)$, in the direction of the *x*-axis $c(dy = 0)$, in the direction of the *y*-axis $c(dx = 0)$, and the extremal values $c_{min}$ and $c_{max}$.

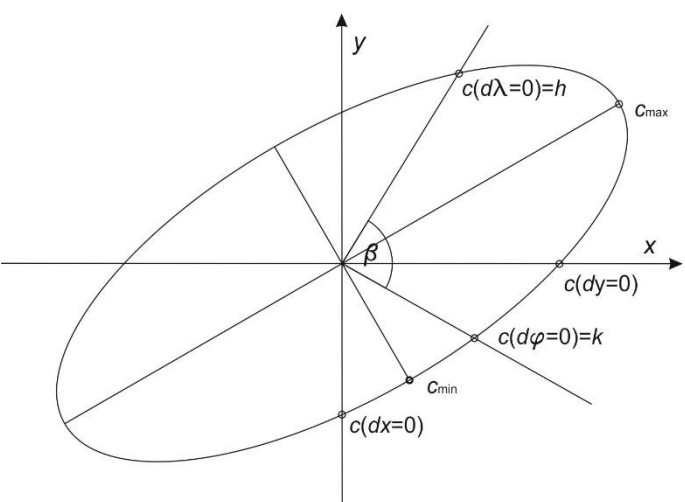

**Figure 3.** General case of Tissot's indicatrix showing local linear scale factors along a meridian $h = c(d\lambda = 0)$, along a parallel $k = c(d\varphi = 0)$, in the direction of the x-axis $c(dy = 0)$, in the direction of the y-axis $c(dx = 0)$, and extremal values $c_{min}$ and $c_{max}$. The angle $\beta$ is the angle between the images of a meridian and a parallel at a point under consideration.

## 4. Local Linear Scale Factor in a Given Direction

Let us suppose that we need a local linear scale factor in a given direction. If the direction is defined by $d\varphi$ and $d\lambda$, we can use Equation (9), and our problem will be solved. If the direction is defined by $dx$ and $dy$, we can use the following procedure. Let us denote that $\psi$ is known as a meridian convergence:

$$\tan \psi = \frac{dy}{dx}. \tag{24}$$

Then:

$$\cos \psi dy = \sin \psi dx \tag{25}$$

and by using (15), (9) can be transformed into:

$$c^2 = \frac{H^2\left(dx^2+dy^2\right)}{M^2\left(\frac{\partial y}{\partial \lambda}dx - \frac{\partial x}{\partial \lambda}dy\right)^2 + N^2\cos^2\varphi\left(\frac{\partial y}{\partial \varphi}dx - \frac{\partial x}{\partial \varphi}dy\right)^2} = \frac{H^2}{a_1\cos^2\psi + a_2\sin\psi\cos\psi + a_3\sin^2\psi^2}, \tag{26}$$

where:

$$\begin{aligned}
a_1 &= M^2\left(\frac{\partial y}{\partial \lambda}\right)^2 + N^2\cos^2\varphi\left(\frac{\partial y}{\partial \varphi}\right)^2, \\
a_2 &= -2\left(M^2\frac{\partial x}{\partial \lambda}\frac{\partial y}{\partial \lambda} + N^2\cos^2\varphi\frac{\partial x}{\partial \varphi}\frac{\partial y}{\partial \varphi}\right), \\
a_3 &= M^2\left(\frac{\partial x}{\partial \lambda}\right)^2 + N^2\cos^2\varphi\left(\frac{\partial x}{\partial \varphi}\right)^2.
\end{aligned} \tag{27}$$

It follows that a local linear scale factor $c$ in the direction $\psi$ defined by (24) can be calculated by the formula:

$$c = \frac{H}{\sqrt{a_1\cos^2\psi + a_2\sin\psi\cos\psi + a_3\sin^2\psi}}, \tag{28}$$

where the coefficients $a_1$, $a_2$, and $a_3$ are given by (27) and $H$ by (16).

In a special case, when $dy = 0$, then $\psi = 0$, and (28) reduces to (29):

$$c(dy = 0) = \frac{H}{\sqrt{a_1}}, \tag{29}$$

which is equivalent to (20).

If $dx = 0$, then $\psi = \frac{\pi}{2}$, and (28) takes the form of (30):

$$c(dx = 0) = \frac{H}{\sqrt{a_3}}, \tag{30}$$

which is equivalent to (23).

The extremal values of $c = c(\psi)$ given by (28) can be obtained in the usual way:

$$\frac{dc}{d\psi} = 0 \tag{31}$$

which gives:

$$\tan 2\psi = \frac{a_2}{a_1 - a_3} \tag{32}$$

and by substituting (27) in (26), the extremal values of $c$ are as follows:

$$\begin{aligned}
c_{1,2} &= \frac{H\sqrt{2}}{\sqrt{a_1+a_3\mp\sqrt{(a_1-a_3)^2+a_2^2}}} = \frac{\sqrt{a_1+a_3\pm\sqrt{(a_1-a_3)^2+a_2^2}}}{\sqrt{2}MN\cos\varphi} = \\
&= \frac{\sqrt{M^2G+EN^2\cos^2\varphi\pm\sqrt{(M^2G+EN^2\cos^2\varphi)^2-4H^2M^2N^2\cos^2\varphi}}}{\sqrt{2}MN\cos\varphi} = \\
&= \frac{\sqrt{h^2+k^2\pm\sqrt{(h^2-k^2)^2+4h^2k^2\cos^2\beta}}}{\sqrt{2}}.
\end{aligned} \tag{33}$$

In (33), $c_{1,2} = h$ and $k$ are the local linear scale factors along a meridian and a parallel, respectively (see (12) and (13)).

The angle $\beta$ (Figure 1) is the angle between the images of a meridian and a parallel at one point (see [10]):

$$\sin^2\beta = \frac{H^2}{EG}, \quad \cos^2\beta = \frac{F^2}{EG}. \tag{34}$$

The extremal values of the local linear scale factor (33) are the semiaxes of the Tissot indicatrix or the ellipse of distortion. In order to be able to visualize an ellipse of distortions,

in addition to its size, the direction of its axes is also required. The formula is known from trigonometry:

$$\tan 2\psi = \frac{2 \tan \psi}{1 - \tan^2 \psi}. \tag{35}$$

If we express $\tan \psi$ from that formula, we obtain the following quadratic equation:

$$\tan^2 \psi + \frac{2}{\tan 2\psi} \tan \psi - 1 = 0. \tag{36}$$

From (36), we see that $\tan \psi_1 \tan \psi_2 = -1$ holds true, which means that the directions $\psi_1$ and $\psi_2$ will be orthogonal.

Relation (33) is not sufficient to determine the direction of the axis of the ellipse because it gives the direction of both axes since $\tan 2\psi = \tan(\pi - 2\psi) = \tan 2\left(\frac{\pi}{2} - \psi\right)$. So, we will solve Equation (36). The solutions of this quadratic equation are as follows:

$$\tan \psi_{1,2} = -\cot 2\psi \pm \sqrt{\cot^2 2\psi + 1}, \tag{37}$$

i.e.,

$$\tan \psi_{1,2} = \frac{a_3 - a_1 \pm \sqrt{(a_3 - a_1)^2 + a_2^2}}{a_2}. \tag{38}$$

The question of which of the directions $\psi_1$ and $\psi_2$ corresponds to the major and which to the minor axis of the ellipse remains open. The larger semiaxis is calculated as:

$$c_1 = \frac{H\sqrt{2}}{\sqrt{a_1 + a_3 - \sqrt{(a_1 - a_3)^2 + a_2^2}}} = \frac{\sqrt{a_1 + a_3 + \sqrt{(a_1 - a_3)^2 + a_2^2}}}{\sqrt{2}MN \cos \varphi} \tag{39}$$

and the corresponding direction as:

$$\tan \psi_1 = \frac{a_3 - a_1 + \sqrt{(a_3 - a_1)^2 + a_2^2}}{a_2}. \tag{40}$$

The smaller semiaxis is as follows:

$$c_2 = \frac{H\sqrt{2}}{\sqrt{a_1 + a_3 + \sqrt{(a_1 - a_3)^2 + a_2^2}}} = \frac{\sqrt{a_1 + a_3 - \sqrt{(a_1 - a_3)^2 + a_2^2}}}{\sqrt{2}MN \cos \varphi} \tag{41}$$

with the corresponding direction:

$$\tan \psi_2 = \frac{a_3 - a_1 - \sqrt{(a_3 - a_1)^2 + a_2^2}}{a_2} \text{ or } \psi_2 = \psi_1 + \frac{\pi}{2}. \tag{42}$$

We can now write Equation (28) in the following form:

$$c(\psi) = \frac{c_1 c_2}{\sqrt{c_1^2 \cos^2(\psi - \psi_1) + c_2^2 \sin^2(\psi - \psi_1)}}, \tag{43}$$

which is the equation of an ellipse in the polar coordinate system. In parametric form in the coordinate system $x, y$ will be:

$$x = c(\psi) \sin(\psi - \psi_1), \; y = c(\psi) \cos(\psi - \psi_1) \tag{44}$$

and in canonical form:

$$\frac{x^2}{c_1^2} + \frac{y^2}{c_2^2} = 1. \tag{45}$$

## 5. Examples

All derived formulas in the previous sections will be applied to several map projections to illustrate their functionality and validity.

### 5.1. Local Linear Scale Factors in the Mercator Projection

The equations for the normal aspect conformal cylindrical or Mercator projection of an ellipsoid are as follows (according to [2], but in different notation):

$$x = n(\lambda - \lambda_0), \; y = n\left[\tanh^{-1}(\sin\varphi) - e\tanh^{-1}(e\sin\varphi)\right] \tag{46}$$

where $\lambda_0 \in [-\pi, \pi]$ represents the longitude of the central meridian, $n > 0$, and $e$ is defined by (2). From (46) we can obtain partial derivatives:

$$\frac{\partial x}{\partial \lambda} = n, \; \frac{\partial x}{\partial \varphi} = 0, \; \frac{\partial y}{\partial \lambda} = 0, \; \frac{\partial y}{\partial \varphi} = \frac{nM}{N\cos\varphi} \tag{47}$$

and then:

$$H = \frac{Mn^2}{N\cos\varphi}, \; a_1 = a_3 = M^2n^2, \; a_2 = 0$$

and by using Formulae (12), (13), (29), and (30):

$$c(\varphi) = c(d\lambda = 0) = c(d\varphi = 0) = c(dx = 0) = c(dy = 0) = \frac{n}{N\cos\varphi} \tag{48}$$

as expected, because the Mercator projection is conformal.

Tissot's indicatrices are circles (Figure 4) whose radii do not depend on the longitude $\lambda$ and increase with latitude $\varphi$ or with the ordinate $y = y(\varphi)$. The standard parallel corresponds to the latitude for which $N\cos\varphi = n$. For example, if $n = a$, then the equator is the standard parallel.

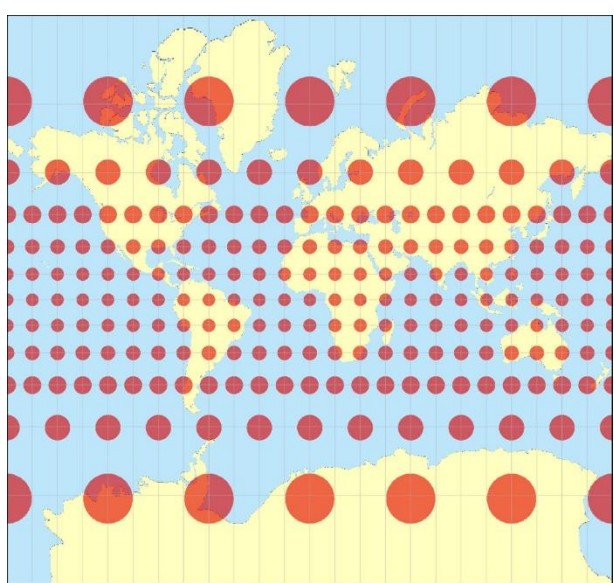

**Figure 4.** Mercator projection map of the earth's sphere with Tissot's indicatrices. Source [11].

*5.2. Local Linear Scale Factors in the Transverse Mercator Projection*

The equations for the transverse aspect conformal cylindrical or transverse Mercator projection of a rotational ellipsoid are [12] as follows:

$$x = lN\cos\varphi + \frac{l^3}{6}N\cos^3\varphi\left(1 - \tan^2\varphi\right) + \cdots$$

$$y = S_\varphi + \frac{l^2}{2}N\sin\varphi\cos\varphi + \cdots, \tag{49}$$

where $l = \lambda - \lambda_0$, $\lambda_0$ represents the longitude of the central meridian, and $S_\varphi$ is the length of the meridian arc from the equator to the point in question. From (49), we can obtain the following partial derivatives:

$$\frac{\partial x}{\partial\lambda} = N\cos\varphi\left(1 + \frac{l^2}{2}\cos 2\varphi\right), \frac{\partial y}{\partial\lambda} = lN\sin\varphi\cos\varphi. \tag{50}$$

The transverse Mercator projection is a conformal projection, and the Cauchy-Riemann conditions for mapping the ellipsoid read as follows:

$$\frac{\partial x}{\partial\varphi} = \frac{M}{N\cos\varphi}\frac{\partial y}{\partial\lambda}, \frac{\partial y}{\partial\varphi} = -\frac{M}{N\cos\varphi}\frac{\partial x}{\partial\lambda} \tag{51}$$

and then:

$$H = MN\cos\varphi\left(1 + l^2\cos^2\varphi + \frac{l^4}{4}\cos^2 2\varphi\right) \approx MN\cos\varphi\left(1 + \frac{l^2}{2}\cos^2\varphi\right)^2,$$

$$a_1 = a_3 = M^2N^2\cos^2\varphi\left(1 + l^2\cos^2\varphi + \frac{l^4}{4}\cos^2 2\varphi\right) = M^2N^2\cos^2\varphi\left(1 + \frac{l^2}{2}\cos^2\varphi\right)^2,$$

$$a_2 = 0$$

and then by using Formulae (12), (13), (29), and (30):

$$c(\varphi, \lambda) = c(d\lambda = 0) = c(d\varphi = 0) = c(dx = 0) = c(dy = 0) \approx 1 + \frac{l^2}{2}\cos^2\varphi \tag{52}$$

as expected because the transverse Mercator projection is also conformal. Tissot's indicatrices are circles (Figure 5) whose radii increase with the longitude $\lambda$. From (52), we see that the local linear scale factor will be equal to 1 for $l = 0$. Therefore, the central meridian of the mapping area is the standard meridian.

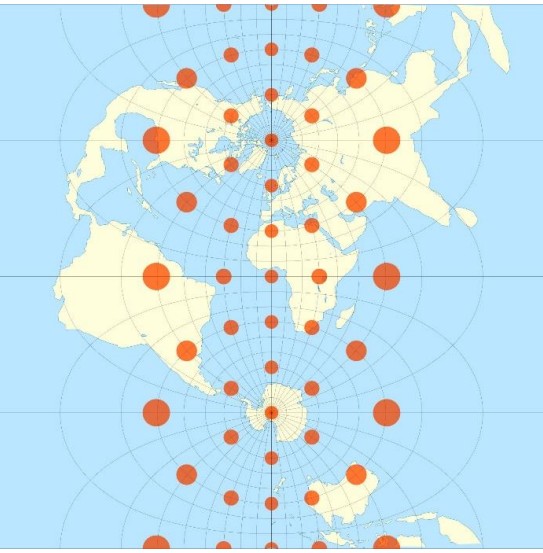

**Figure 5.** The world on a transverse Mercator projection of the Earth's sphere, with 10° graticule and Tissot's indicatrices overlaid. Source [13].

### 5.3. Local Linear Scale Factors in the Web-Mercator Projection

The equations for the Web-Mercator projection of an ellipsoid are [14] as follows:

$$x = a\lambda, \; y = a\tanh^{-1}(\sin\varphi), \tag{53}$$

where $\varphi$ and $\lambda$ are the ellipsoidal latitude and longitude, respectively. From (53), we can obtain partial derivatives:

$$\frac{\partial x}{\partial \lambda} = a, \; \frac{\partial x}{\partial \varphi} = 0, \; \frac{\partial y}{\partial \lambda} = 0, \; \frac{\partial y}{\partial \varphi} = \frac{a}{\cos\varphi} \tag{54}$$

and then:

$$E = \frac{a^2}{\cos^2\varphi}, \; F = 0, \; G = a^2, \; H = \frac{a^2}{\cos\varphi} \tag{55}$$

$$a_1 = N^2 a^2, \; a_2 = 0, \; a_3 = M^2 a^2.$$

By using Formulae (12), (13), (29), and (30), we have the following:

$$h = h(\varphi) = c(d\lambda = 0) = \frac{\sqrt{E}}{M} = \frac{a}{M\cos\varphi} = \frac{\sqrt{\left(1 - e^2\sin^2\varphi\right)^3}}{(1 - e^2)\cos\varphi} \tag{56}$$

$$k = k(\varphi) = c(d\varphi = 0) = \frac{\sqrt{G}}{N\cos\varphi} = \frac{a}{N\cos\varphi} = \frac{\sqrt{1 - e^2\sin^2\varphi}}{\cos\varphi} \tag{57}$$

$$c(dx = 0) = \frac{a}{M\cos\varphi} = h \tag{58}$$

$$c(dy = 0) = \frac{a}{N\cos\varphi} = k \tag{59}$$

Furthermore:

$$\frac{h}{k} = \frac{1 - e^2\sin^2\varphi}{1 - e^2} \tag{60}$$

and

$$\left(\frac{h}{k}\right)_{max} = \frac{1}{1 - e^2} \approx 1 + e^2 \tag{61}$$

which is achieved for $\varphi = 0$. The Web-Mercator projection is not conformal. Tissot's indicatrices are ellipses very close to circles. Their semiaxes do not depend on the longitude, and the largest ratio of these semiaxes is at the equator and is approximately $1 + e^2 = 1.0067$. There is no standard parallel, i.e., a parallel along which $h = k = 1$ would be valid [14]. The distribution of distortions is very similar to that of the Mercator projection (Figure 4). The difference cannot be seen on a small-scale map.

### 5.4. Local Linear Scale Factors in the Albers Equal-Area Conic Projection

The equations for the normal aspect Albers or equal-area conic projection of an ellipsoid are the following [2]:

$$x = \rho\sin\theta, \; y = \rho_0 - \rho\cos\theta \tag{62}$$

where $\varphi$ and $\lambda$ are the ellipsoidal latitude and longitude, respectively, calculated as follows:

$$\rho = \frac{a}{n}\sqrt{C - nq(\varphi)}, \; \theta = n(\lambda - \lambda_0) \tag{63}$$

$$q(\varphi) = \left(1 - e^2\right)\left(\frac{\sin\varphi}{1 - e^2\sin^2\varphi} - \frac{1}{2e}\ln\frac{1 - e\sin\varphi}{1 + e\sin\varphi}\right)$$

$$\rho_0 = \frac{a}{n} \sqrt{C - nq(\varphi_0)}$$

$$m(\varphi) = \frac{\cos \varphi}{\sqrt{1 - e^2 \sin^2 \varphi}}$$

$$C = m^2(\varphi_1) + nq(\varphi_1), n = \frac{m^2(\varphi_1) - m^2(\varphi_2)}{q(\varphi_2) - q(\varphi_1)}$$

where $\varphi_0$ and $\lambda_0$ are the latitude and longitude, respectively, for the origin of the rectangular coordinates, and $\varphi_1$ and $\varphi_2$ are the latitudes of standard parallels.

From (62) and (63), we can obtain the following partial derivatives:

$$\frac{\partial x}{\partial \lambda} = n\rho \cos n(\lambda - \lambda_0), \ \frac{\partial x}{\partial \varphi} = -M\frac{r}{n\rho} \sin n(\lambda - \lambda_0),$$

$$\frac{\partial y}{\partial \lambda} = n\rho \sin n(\lambda - \lambda_0), \ \frac{\partial y}{\partial \varphi} = M\frac{r}{n\rho} \cos n(\lambda - \lambda_0), \tag{64}$$

and then:

$$a_1 = \frac{M^2}{n^2\rho^2} \left[ n^4\rho^4 \sin^2 n(\lambda - \lambda_0) + r^4 \cos^2 n(\lambda - \lambda_0) \right]$$

$$a_2 = -2\frac{M^2}{n^2\rho^2} \left( n^4\rho^4 - r^4 \right) \sin n(\lambda - \lambda_0) \cos n(\lambda - \lambda_0)$$

$$a_3 = \frac{M^2}{n^2\rho^2} \left[ n^4\rho^4 \cos^2 n(\lambda - \lambda_0) + r^4 \sin^2 n(\lambda - \lambda_0) \right],$$

and then, by using Formulae (12), (13), (29), and (30), we can obtain:

$$h = c(d\lambda = 0) = \frac{1}{k} = \frac{m}{\sqrt{C - nq}},$$

$$k = c(d\varphi = 0) = \frac{n\rho}{r} = \frac{\sqrt{C - nq}}{m} \tag{65}$$

$$c(dy = 0) = \frac{nr\rho}{\sqrt{n^4\rho^4 \sin^2 n(\lambda - \lambda_0) + r^4 \cos^2 n(\lambda - \lambda_0)}},$$

$$c(dx = 0) = \frac{nr\rho}{\sqrt{\sqrt{n^4\rho^4 \cos^2 n(\lambda - \lambda_0) + r^4 \sin^2 n(\lambda - \lambda_0)}}}. \tag{66}$$

According to (39) and (41), the larger and smaller semiaxes of the Tissot's indicatrices are:

$$c_1 = c_1(\varphi) = \frac{n\rho}{r} = \frac{a\sqrt{C - nq(\varphi)}}{N \cos \varphi} = k, \ c_2 = c_2(\varphi) = \frac{r}{n\rho} = \frac{N \cos \varphi}{a\sqrt{C - nq(\varphi)}} = h.$$

We can see that the local linear scale factors (65)–(66) are different and depend on the direction. The distribution of distortions is depicted in Figure 6, although the choice of standard parallels for the map in that Figure is not in accordance with the theory [10,15].

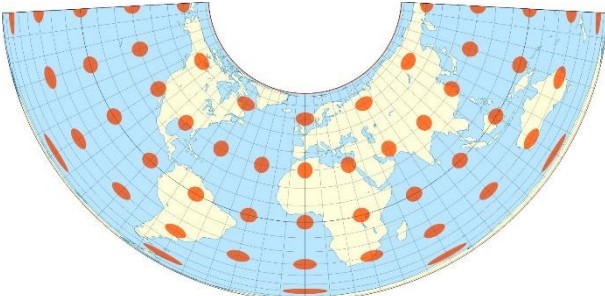

**Figure 6.** The world on an Albers projection of the Earth's sphere, with 10° graticule and Tissot's indicatrices overlaid. Standard parallels are at 45° N and 15° N. Source [16].

## 6. Instead of Conclusions

All map projections involve distortion of areas, angles, and/or distances. The types of distortion can be controlled to preserve specific characteristics, but map projections must distort other characteristics of the represented object. The main problem in cartography is that it is not possible to map/project/transform a spherical or ellipsoidal surface into a plane with zero distortions.

Noting distortions is not enough. Cartographers need a way to measure the amount of distortion in a given projection. Some specific computational procedure must be followed. Such a procedure, typically described by a mathematical formula, is called the *distortion measure*. Therefore, in order to evaluate any kind of distortions, we need some kind of measure. There is no perfect choice because every map projection uniquely alters some aspect of space during the transformation process. Detailed information about the type, amount, and distribution of distortion is essential for choosing the best projection for a particular map or data set [17]. The distortion inherent in projections can be measured and symbolized much like any other map variable [18]. Map projections and the distortions they carry need to be better understood by spatial data developers, distributors, and users. Map distortions should be included along with map data as confidence layers, and easily accessible distortion displays should be available to help in the selection of map projections.

It is well known that scale changes from point to point, and at certain points, usually depends on direction. This is the local scale. The local linear scale factor $c$ is the ratio of the differential of the curve arc in the plane of projection and the differential of the corresponding curve arc on an ellipsoid or spherical surface. The local linear scale factor $c$ is one of the most important indicators of distortion distribution in the theory of map projections.

Knowing the distribution of distortion is important when choosing a map projection [19]. In any software that serves to select a map projection, as well as in any GIS, distortion display options should be built in. Formulae for determining the local linear scale factor in a direction of a coordinate axis are especially important in working with raster data [8,20–22].

It is not possible to read the local linear scale factor in a direction of a coordinate axis immediately from the definition (9). The same is true for Equation (10), where $\alpha$ is the angle between the meridian and any direction in a point in question. In the paper, we derive new formulae that enable calculation of a local linear scale factor in the direction of coordinate axes $x$ and $y$ for a rotational ellipsoid. Moreover, we derive the formula for computing the local linear scale factor in any direction defined by $dx$ and $dy$. The position and magnitude of the extreme values of the local linear scale factor were also considered, and new formulae were derived for a rotational ellipsoid.

**Funding:** This research received no external funding.

**Data Availability Statement:** Data sharing is not applicable to this article as no new data were created or analyzed in this study.

**Conflicts of Interest:** The author declares no conflict of interest.

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
