# Peer review of "Local Linear Scale Factors in Map Projections of an Ellipsoid"

_geographies, doi:10.3390/geographies1030014_

Round 1

Reviewer 1 Report

This is a well written and mathematically clearly deduced piece of research on local linear scale factors. My recommendations for the final publications are minor ones, namely:

Given the wide scope of topics in our journal and the wide interests of its readers, a broader introduction into the problem the paper builds upon could be useful. For example, you could firstly explain the concept of linear scale and afterwards introduce the term of a scale factor to express a ratio (cf. Bugayevskiy & Snyder), not just in the conclusion. Also mentioning the idea of Tissot's indicatrix already section one could help to visualize the problem.

At the end of the paper and again for a rather general audience, it would be interesting to read about practical consequences of these new formulae? Need/Will these formulae get integrated into GIS software? Will they facilitate a better understanding at to moment of choosing/arguing a map projection? The relevance of your findings may be difficult to understand for many readers, so a few additional lines on this regard might be appreciated.

In line 192, you mention “Thomas 1952”, which seems to be a reference not mentioned in the bibliography?

Author Response

My comments are in red.

This is a well written and mathematically clearly deduced piece of research on local linear scale factors. My recommendations for the final publications are minor ones, namely:

Given the wide scope of topics in our journal and the wide interests of its readers, a broader introduction into the problem the paper builds upon could be useful. For example, you could firstly explain the concept of linear scale and afterwards introduce the term of a scale factor to express a ratio (cf. Bugayevskiy & Snyder), not just in the conclusion. Also mentioning the idea of Tissot's indicatrix already section one could help to visualize the problem.

The remark is accepted, a broader introduction is included.

At the end of the paper and again for a rather general audience, it would be interesting to read about practical consequences of these new formulae? Need/Will these formulae get integrated into GIS software? Will they facilitate a better understanding at to moment of choosing/arguing a map projection? The relevance of your findings may be difficult to understand for many readers, so a few additional lines on this regard might be appreciated.

The remark is accepted, s few additional lines is included in the last section titled „Instead of conclusion“.

In line 192, you mention “Thomas 1952”, which seems to be a reference not mentioned in the bibliography?

The remark is accepted, too. I am sorry for the mistake.

Reviewer 2 Report

This manuscript presents a method to compute the Local Linear Scale Factors in Map Projections of an Ellipsoid. It is based on the general map projection formula and distortion circle computation. The study selects the x direction or y direction and computes the local linear scale factor. The main works is based on the reference 2 and 3.

  • In the introduction, the related background works should be discussed. More texts need to add about the distortion representation in map projection, the main methods in scale representation, the comparison between different methods and others. The related works is very traditional in cartography field. The author need to find more reference to add to describe questions above.
  • In section 2, before present formula, the background information needs to add, and a figure of ellipsoid and map projection is required to insert. Based on the figure describes geometric parameter and scale.
  • The discussion of scale computation needs to add, including the application fields and support condition.
  • As for the computation method, the software and platform needs to describe

Author Response

My comments are in red.

This manuscript presents a method to compute the Local Linear Scale Factors in Map Projections of an Ellipsoid. It is based on the general map projection formula and distortion circle computation. The study selects the x direction or y direction and computes the local linear scale factor. The main works is based on the reference 2 and 3.

It is not about circles, but about ellipses of distortion.

  • In the introduction, the related background works should be discussed. More texts need to add about the distortion representation in map projection, the main methods in scale representation, the comparison between different methods and others. The related works is very traditional in cartography field. The author need to find more reference to add to describe questions above.

The remark is accepted, a broader introduction is included, as well as more references cited.

  • In section 2, before present formula, the background information needs to add, and a figure of ellipsoid and map projection is required to insert. Based on the figure describes geometric parameter and scale.

The required figure is inserted, while the background information is added in the introductory section.

  • The discussion of scale computation needs to add, including the application fields and support condition.

The discussion on scale computation is not added, because there is nothing special regarding the scale computation. One needs to apply only basic mathematical operations to compute the scale.

  • As for the computation method, the software and platform needs to describe

Although there are many equations in the paper, there is no need to describe the software and platform for the computation method, because the needed computations can be performed by using e.g. Excel, or any simple calculator.

Round 2

Reviewer 2 Report

The revised manuscript has added some contents in the introduction to enhance the research background. The required figure has added to illustrate the basic distoration ideas. The further consideration about the manuscript can be  (1)  There are a lot of formulas in the text. Which one is directly referenced from other materials? which one is extended based on other formula. These points can be added with references. (2) The figures need to enhance to guarantee the resolution.

Author Response

(1) There are a lot of formulas in the text. Which one is directly referenced from other materials? which one is extended based on other formula. These points can be added with references.

The remark is accepted. The needed reference numbers are included. I used track changes to make them visible.

(2) The figures need to enhance to guarantee the resolution.

The remark is accepted. All figures are changed, and they are in 300 dpi now.